# Left Ventricular Morphology and Function as a Determinant of Pulmonary Hypertension in Patients with Severe Aortic Stenosis: Cardiovascular Magnetic Resonance Imaging Study

**DOI:** 10.3390/medicina55100711

**Published:** 2019-10-22

**Authors:** Birute Gumauskiene, Lina Padervinskiene, Jolanta Justina Vaskelyte, Audrone Vaitiekiene, Tomas Lapinskas, Deimante Hoppenot, Skaidrius Miliauskas, Gryte Galnaitiene, Paulius Simkus, Egle Ereminiene

**Affiliations:** 1Department of Cardiology, Medical Academy, Lithuanian University of Health Sciences, LT 44307 Kaunas, LithuaniaAudrone.Vaitiekiene@lsmuni.lt (A.V.); Tomas.Lapinskas@lsmuni.lt (T.L.); eglerem@yahoo.com (E.E.); 2Department of Radiology, Medical Academy, Lithuanian University of Health Sciences, LT 44307 Kaunas, Lithuania; padervinskiene@gmail.com (L.P.); gryte.galnaitiene@gmail.com (G.G.); pauliusimkus@gmail.com (P.S.); 3Department of Pulmonology, Medical Academy, Lithuanian University of Health Sciences, LT 44307 Kaunas, Lithuania; deimante@gmail.com (D.H.); skaidrius.Miliauskas@lsmuni.lt (S.M.)

**Keywords:** aortic stenosis, pulmonary hypertension, left ventricular, cardiovascular magnetic resonance, feature tracking

## Abstract

*Background and Objectives:* The influence of cardiac magnetic resonance (CMR) derived left ventricular (LV) parameters on the prognosis of patients with aortic stenosis (AS) was analyzed in several studies. However, the data on the relations between the LV parameters and the development of pulmonary hypertension (PH) in severe AS is lacking. Our objectives were to evaluate the CMR-derived changes of the LV size, morphology, and function in patients with isolated severe AS and PH, and to investigate the prognostic impact of these parameters on elevated systolic pulmonary artery pressure (sPAP). *Materials and Methods:* Thirty patients with isolated severe AS (aortic valve area ≤1 cm^2^) underwent a 2D-echocardiography (2D echo) and CMR before aortic valve replacement. Indices of the LV mass and volumes and ejection fraction were analyzed by CMR. The LV global longitudinal (LV LGS) and circumferential strain (LV CS) were calculated using CMR feature tracking (CMR-FT) software (Medis Suite QStrain 2.0, Medis Medical Imaging Systems B.V., Leiden, The Netherlands). The LV fibrosis expansion was assessed using a late gadolinium enhancement sequence. PH was defined as having an estimated sPAP of ≥45 mm Hg. The statistical analysis as performed using SPSS version 23.0 (SPSS, Chicago, IL, USA) *Results:* 30 patients with severe AS were included in the study, 23% with severe isolated AS had PH (mean sPAP 55 ± 6.6 mm Hg). More severe LV anatomical and functional abnormalities were observed in patients with PH when compared with patients without PH—a higher LV end-diastolic volume index (EDVi) (140 [120.0–160.0] vs. 90.0 mL/m² [82.5–103.0], *p* = 0.04), larger LV fibrosis area (7.8 [5.6–8.0] vs. 1.3% [1.2–1.5], *p* = 0.005), as well as lower LV global longitudinal strain (GLS; −14.0 [−14.9–(−8.9)] vs. −21.1% [−23.4–(−17.8)], *p* = 0.004). By receiver–operating characteristic (ROC) curve analysis, LV EDVi > 107.7 mL/m² (Area Under the Curve (AUC) 95.7%), LV GLS < −15.5% (AUC 86.3%), and LV fibrosis area >5% (AUC 89.3) were found to be robust predictors of PH in severe AS patients. *Conclusions:* In patients with severe aortic stenosis, a larger end-diastolic LV volume, impaired LV global longitudinal strain, and larger LV fibrosis extent can predict the development of pulmonary hypertension.

## 1. Introduction

Pulmonary hypertension (PH) is related to poor outcomes regardless of surgical or interventional treatment in aortic stenosis (AS) patients [1,2,3]. Therefore, an evaluation of the risk factors for the development of PH in this group of patients is of great importance to improve risk stratification. Several studies have evaluated the echocardiographic predictors for the genesis of PH in AS [4,5]. Nevertheless, data about the impact of cardiac magnetic resonance (CMR) derived parameters are still lacking. Modern CMR imaging techniques may provide accurate measurements of cardiac chambers, as well as about the progression of myocardial fibrosis, and, combined with feature tracking (CMR-FT), helps to detect early subclinical ventricular dysfunction [6,7,8,9]. A large number of studies using CMR investigated a variety of left (LV) and right ventricular (RV) structural and functional parameters in pre-capillary PH patients, as well as their prognostic significance [10,11,12]. However, there is still scarce information about the LV structure and function in PH with AS. Thus we aimed to evaluate the LV size, function and mechanics in patients with isolated severe AS and PH using novel CMR technique as well as to investigate the impact of these parameters on elevated pulmonary systolic pressure (sPAP).

## 2. Materials and Methods

### 2.1. Study Population

The study included 30 patients with symptomatic severe AS, defined as an aortic valve area less than or equal to 1 cm², who underwent an 2D-echocardiography (2D echo) and CMR evaluation before aortic valve replacement surgery. The patients were divided into two groups according to the presence or absence of PH, with an sPAP cut-off value of 45 mm Hg derived by 2D echo. The patients with documented coronary heart disease, chronic pulmonary disease, atrial fibrillation, and moderate or severe mitral regurgitation were excluded from the study.

The study was approved by the Kaunas Regional Biomedical Research Ethics Committee, no. BE-2-8, issued on 29 October 2014. All of the patients gave their informed written consent.

### 2.2. Transthoracic Echocardiography

The 2D echo was performed using a GE Vivid 7 system (GE Vingmed Ultrasound AS N-3190, Horten, Norway), by an independent, experienced echocardiographer. Digital loops were stored and analyzed offline (EchoPac V.6.0.0; GE Vingmed, Chalfont St. Giles, United Kingdom). 

A conventional 2D echo was performed from the left parasternal and apical windows. Echocardiographic LV and aortic valve area measurements were performed according to the American Society of Echocardiography recommendations [13]. PH was defined as an estimated sPAP of ≥45 mm Hg, based on a tricuspid regurgitation flow velocity of ≥3.0 m/s and on the presence of other echocardiographic PH signs [14]. 

### 2.3. Cardiac Magnetic Resonance Imaging Measurements

CMR was performed using a 1.5T whole-body system (Siemens Aera, Siemens Medical Solutions; Erlangen, Germany). The CMR protocol consisted of an ECG gated cine sequence obtained during a short breath-hold. The cine views were obtained in three LV long (two, three, and four chambers) and short axis views. The end-diastole and end-systole of the LV were defined as the maximum and minimum volumes on balanced steady-state free precession (bSSFP) sequences. 

The end-diastolic volume (EDV), end-systolic volume (ESV), myocardial mass (MM), and ejection fraction (EF) of LV were calculated in standard cine images using an MR analysis software system (syngo.via; Siemens Healthcare, Erlangen, Germany. The indices of the LV volumes (EDVi and ESVi) and myocardial mass index (MMi) were calculated based on the body surface area. 

The LV fibrosis expansion was assessed using a late gadolinium enhancement (LGE) sequence. The LGE images were obtained 10 minutes after the infusion of gadolinium (0.2 mmol/kg). The LV fibrosis extent was assessed using a CMR analysis software system (syngo.via; Siemens Healthcare). To assess the midwall LGE quantitatively, short-axis slices were inspected visually, the area of LGE was traced manually, and the fibrosis area results were expressed as a percentage of the myocardial mass.

### 2.4. Feature Tracking Analysis

The CMR images were analyzed by two experienced radiologists using a commercial FT software package (Medis Suite QStrain 2.0; Medis Medical Imaging Systems B.V., Leiden, The Netherlands). The endocardial contours of LV end-diastole and end-systole were marked semi-automatically throughout the cardiac cycle on standard CMR balanced steady-state free precession (bSSFP) sequences. The contours were manually corrected, if necessary. The LV global longitudinal strain (GLS) was calculated by averaging the strain curves of two-, three-, and four-chamber long-axis views. The LV global circumferential strain (GCS) was calculated by averaging the strain curves of the basal, mid, and apical segments obtained from the short-axis views. 

### 2.5. Statistical Analysis

The continuous variables were expressed as the median (interquartile range), and the categorical variables were described as a percentage (number). The continuous variables distribution was verified using the Shapiro–Wilk test. The Mann–Whitney test was used to compare the quantitative sizes of two independent samples. The correlations were computed using Spearman’s method. Chi-square (Fisher’s Exact Test) tests were used to compare the frequencies of the qualitative variables. The results were considered statistically significant when the two-tailed *p*-value was <0.05.

The incremental value of the CMR variable in predicting the development of sPAP ≥ 45 mm Hg was assessed in terms of the construction of the receiver–operating characteristic (ROC) curves. All of the statistical analyses were performed with SPSS 23.0 software (SDSPSS, Chicago, IL, USA).

## 3. Results

Thirty symptomatic patients (mean age 70 (64–75) years) with severe AS (aortic valve area ≤1 cm²) were prospectively enrolled in the study. Among all of the patients, seven (23%) had PH (mean sPAP 55 ± 6.6 mm Hg). The clinical, echocardiographic, and demographic characteristics of the overall cohort, and those with and without PH are shown in Table 1.

The patients in both groups were similar in terms of age, sex, and body mass index. The groups did not differ in terms of the New York Heart Association (NYHA) functional class and with the prevalence of comorbidities such as arterial hypertension, diabetes mellitus, and renal insufficiency. The aortic valve area did not differ between the groups. 

LV EF, estimated by 2D echo and CMR, did not differ between the groups, and was not related to PH (*p* > 0.05; Table 1 and Table 2). The LV MM and MMi, obtained by CMR, did not differ between the groups, and was also not associated with elevated sPAP (*p* > 0.05). For LV EDV and EDVi, ESV, and ESVi, the LV fibrosis area was significantly larger in the PH group when compared to the AS patients without PH (*p* < 0.05) (Table 2). The LV GLS and GCS was lower in patients with elevated sPAP (*p* = 0.004) (Table 2).

Increased sPAP (≥45 mm Hg) correlated with LV EDVi (*r* = 0.6, *p* < 0.000), LV GLS (*r* = 0.5, *p* = 0.007), and LV fibrosis (*r* = 0.6, *p* = 0.004), however it did not correlate with LV EF (*r* = −0.3, *p* = 0.206) and LV GCS (*r* = 0.4, *p* = 0.052).

The ROC analysis data showed that increased LV EDVi > 107.7 mL/m², severely reduced LV GLS > −14.2% and extended LV fibrosis >5% (with a sensitivity of 100% and a high specificity) may predict the development of pulmonary hypertension in severe AS patients (Table 3, Figure 1).

## 4. Discussion

Our study evaluated the changes of LV size, morphology, function, and mechanics in patients with severe AS, and found that a larger LV end-diastolic volume, impaired LV global longitudinal strain, as well as more significant larger LV fibrosis extent derived by CMR are predictors of the development of PH in severe AS patients. 

Increased afterload in severe AS is the leading cause of LV hypertrophy [15]. All of our patients with severe AS had increased LV MMi (compared with the normal range, according to the guidelines [13]. However, our data demonstrated that increased LV MMi was not related to elevated sPAP (*p* = 0.52), probably because a higher LV mass reflects a normal adaptation to an increase in pressure overload, or maybe our PH study group was too small to reflect changes. 

In the end-stage of the hypertrophic process, LV dilatation can be observed in AS patients [15]. In our study, LV EDV and ESV were significantly increased in the PH group (*p* < 0.001). Lancellotti et al. confirmed that the LV EDV measured by 2D echocardiography could be a predictor of exercise PH in asymptomatic severe AS [4]. However, the data concerning CMR derived LV end-diastolic and end-systolic volumes for PH prediction in severe AS is lacking. We found LV EDVi to be a predictor of PH in severe AS, but it is not an early indicator, as it is found in the late phase of the disease, reflecting the pathological remodeling of the ventricle. 

In our study, patients had a normal or slightly reduced LV EF, while a significant correlation between LV EF and elevated sPAP was not found. This data coincides with other studies demonstrating that the LV systolic function was not also associated with PH in AS [16,17]. However, previous studies showed that LV EF is not an accurate parameter to appreciate systolic impairment in AS, and can even overestimate it in the presence of concentric LV hypertrophy [18]. Therefore, the detection of early LV dysfunction before changes of LV EF is of great importance, and it becomes possible using new imaging techniques such as feature tracking (FT) CMR.

The longitudinal LV global strain is influenced by LV overload, hypertrophic remodeling, and chamber dimensions, and first begins to decline when the LV EF is maintained [19]. Our FT-CMR study determined that the worsening of CMR LV GLS was the major independent predictor of PH in severe AS, and this parameter could be the earliest predictor of sPAP elevation. Other studies using echocardiography have identified similar results, that a reduced LV global longitudinal strain in AS was associated with PH [19].

Previous research studies indicated that LV EF is related mainly to the global radial LV function. When LV EF begins to decrease, a gradual loss of circumferential shortening can be observed [20,21]. Nevertheless, our study showed lower LV GCS within PH group which correlated with LV EF (*r* = 0.6, *p* = 0.01) and LV volume (*r* = −0.7, *p* = 0.01, however it did not correlate with PH, probably mainly due to the small sample size

AS is accompanied by the progressive development of interstitial fibrosis, followed by replacement fibrosis. Late gadolinium enhancement (LGE) is a well-established imaging method to detect myocardial fibrosis in AS [9]. Several studies analyzed that replacement fibrosis detected by LGE is a strong predictor of poor outcome [22,23]. However, the data about the myocardial fibrosis impact on PH development is lacking. We analyzed the extent of replacement fibrosis, and revealed that LV fibrosis was significantly higher in the PH group, and a significant correlation between LV fibrosis and elevated sPAP was determined. 

Larger scale studies using CMR and CMR-FT are needed in severe AS with PH patients, in order to assess the changes of the ventricles size, morphology, function, and their association with elevated sPAP. Evaluation of these risk factors for the development of PH in this group of patients is critical to improve the risk stratification. These findings could help to select the patients who would benefit from earlier surgery before the PH development.

## 5. Conclusions

In patients with severe aortic stenosis, a larger end-diastolic LV volume, impaired LV global longitudinal strain, and larger LV fibrosis extent can predict the development of pulmonary hypertension.

## Figures and Tables

**Figure 1 medicina-55-00711-f001:**
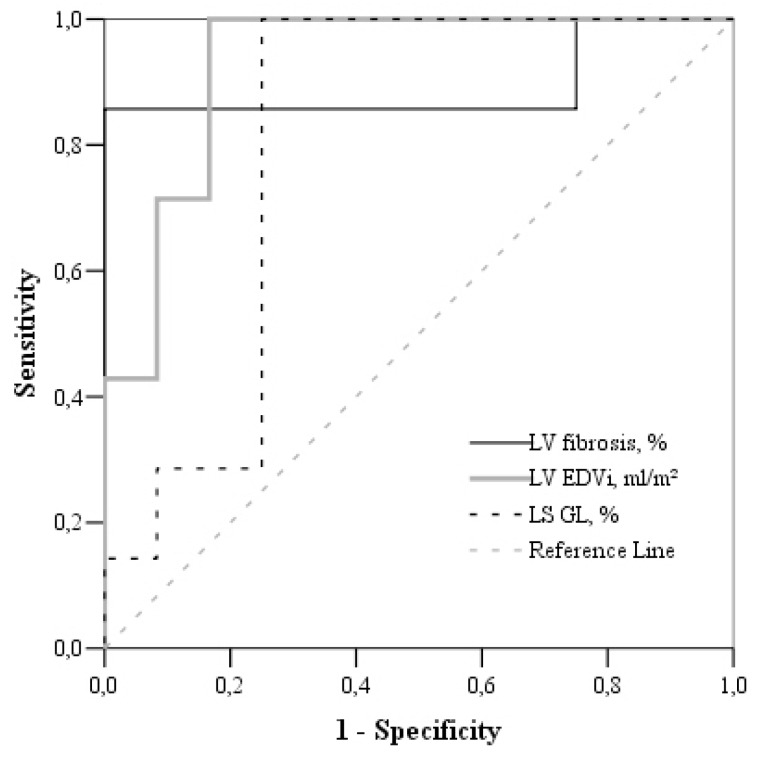
Receiver–operating characteristic (ROC) curve analysis of CMR-derived LV LGS, LV EDVi, and LV fibrosis in the prediction of sPAP ≥ 45 mm Hg.

**Table 1 medicina-55-00711-t001:** Demographic, clinical, and echocardiographic parameters of patient cohort.

Characteristic	Overall (*n* = 30)	sPAP ≥ 45 mm Hg (*n* = 7)	sPAP < 45 mm Hg (*n* = 23)	*p*-value
Age, years	70 (64–75)	73 (60–75)	70 (65–76)	0.883
Sex, male/female, *n* (%)	16 (53.3)/14 (46.7)	4 (57.1)/3 (42.9)	12 (52.2)/11(47.8)	1.0
Arterial hypertension, *n* (%)	25 (83.3)	6 (85.7)	19 (82.3)	1.0
Diabetes mellitus, *n* (%)	2 (6.7)	0	2 (6.7)	1.0
NYHA functional class 2–3, *n* (%)	26 (86.7)	6 (85.7)	20 (87.0)	1.0
Body mass index, kg/m^2^	28.5 (25.0–32.30)	27.0 (22.0–32.0)	29.0 (25.0–33.0)	0.349
Glomerulal filtration rate, mL/min/1.73 ^2^	83.5 (65.0–101.0)	74.0 (58.0–83.0)	87.0 (65.0–106.0)	0.077
Aortic valve area, cm^2^	0.85 (0.64–0.96)	0.87 (0.75–0.95)	0.8 (0.63–1.0)	0.571
Mean aortic gradient (mm Hg)	43 (40–46)	42 (40–45)	43 (40–46)	0.708
sPAP, mm Hg	37.5 (35.5–43.8)	57.0 (49.0–63.0)	36.0 (33.0–41.0)	<0.001
LV end-diastolic diameter index, mm/m^2^	25.5 (22.8–29.3)	30.0 (29.0–32.0)	24.0 (22.0–27.0)	0.003
LV EF (%)	55.0 (50.0–55.0)	50.0 (40.0–55.0)	55.0 (50.0–55.0)	0.809

Values are expressed as number (%) or median (interquartile range); NYHA—New York Heart Association; sPAP—systolic pulmonary arterial pressure; LV—left ventricular; EF—ejection fraction.

**Table 2 medicina-55-00711-t002:** Left ventricular measurements: magnetic resonance imaging and feature tracking parameters.

Parameter	sPAP ≥ 45 mm Hg (*n* = 7)	sPAP < 45 mm Hg (*n* = 23)	*p*-value
LV EDV, mL	289.0 (245.0–337.0)	179.4 (159.0–200.4)	<0.001
LV EDVi, mL/m²	140 (120.0–160.0)	90.0 (82.5–103.0)	<0.001
LV ESV, mL	109 (94–111)	59 (50–84)	<0.001
LV ESVi, mL/m²	55 (55–60)	32 (27–36)	<0.001
LV MM, g	235.0 (180.0–263.0)	197.0 (141.1–306.0)	0.54
LV MMi, g/m^2^	121 (108–148)	103 (82–140)	0.26
LV EF, %	51.5 (42.2–64.3)	61.5 (47.0–68.3)	0.23
LV GLS, %	−14.0 (−14.9–(−8.9))	−21.1 (−23.4–(−17.8))	0.004
LV GCS, %	−16.4 (−15.4–(−10.9))	−32.0 (−36.1–(−20.8))	0.004
LV fibrosis, %	7.8 (5.6–8.0)	1.3 (1.2–1.48)	0.005

LV—left ventricular; ESV—end-systolic volume; ESVi—end-systolic volume index; EDV—end-diastolic volume; EDVi—end-diastolic volume index; MM—myocardial mass; MMi—myocardial mass index; EF—ejection fraction; GLS—global longitudinal strain; GCS—global circumferential strain.

**Table 3 medicina-55-00711-t003:** Receiver–operating characteristic (ROC) analysis data. Cardiac magnetic resonance (CMR) parameters in the prediction of elevated sPAP (≥45 mm Hg).

Parameters/Threshold	Area under Curve (95% CI)	Sensitivity (%)	Specificity (%)
LV EDVi > 107.7 mL/m²	95.7 (88.9–102)	100	87
LV LGS > −15.5%	86.3 (73.0–99.7)	100	82.6
LV fibrosis > 5%	89.3 (69.3–109.3)	100	91.7

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
