# Peer review of "Left Ventricular Morphology and Function as a Determinant of Pulmonary Hypertension in Patients with Severe Aortic Stenosis: Cardiovascular Magnetic Resonance Imaging Study"

_medicina, 2019, doi:10.3390/medicina55100711_

Round 1

Reviewer 1 Report

The authors have satisfactorily responded to all my questions and made the necessary changes to the manuscript.

Author Response

Thank You for your corrections and consideration of this manuscript

Reviewer 2 Report

The study by Gumauskiene et al evaluated the correlation between CMR parameters and pulmonary hypertension measured by echocardiogram. Although the study is original, there are several major limitations:

1) The authors defined severe aortic stenosis as an aortic valve area less than 1,0 cm². However, current guidelines define severe aortic stenosis as aortic valve area less or equal than 1.0 cm².

2) Why transaortic mean gradient was not used? Low-flow low-gradient AS patients were also included?

3) What was the method used to calculate aortic valve area?

4) Table 1: besides severe aortic stenosis definition, it seems that patients with aortic valve area equal than 1.0 cm² were included in sPAP < 45 mmHg group (0.8 [0.63-1.0])

5) What was the LGE pattern?

6) ROC curve is not a prediction tool. This is a major methodological limitation of this study. It is not clear the real contribution of this study, since CMR parameters can only stratify patients with sPAP > 45 mmHg. In other words, is it necessary to undergo CMR just to confirm that the patient has sPAP > 45 mmHg? (lines 137, 139, 147, 158, 170, 193)

7) Correlation between LV fibrosis and elevated sPAP was only moderate (r=0.6) (lines 184-185)

8) What was the definition of coronary heart disease?

9) Sample size was too small for the study design.
